# Coarse-to-Fine Localization of Underwater Acoustic Communication Receivers

**DOI:** 10.3390/s22186968

**Published:** 2022-09-14

**Authors:** Pan He, Lu Shen, Benjamin Henson, Yuriy V. Zakharov

**Affiliations:** Department of Electronic Engineering, University of York, York YO10 5DD, UK

**Keywords:** ambiguity function, matched field processing, receiver localization, refinement, underwater acoustic communications

## Abstract

For underwater acoustic (UWA) communication in sensor networks, the sensing information can only be interpreted meaningfully when the location of the sensor node is known. However, node localization is a challenging problem. Global Navigation Satellite Systems (GNSS) used in terrestrial applications do not work underwater. In this paper, we propose and investigate techniques based on matched field processing for localization of a single-antenna UWA communication receiver relative to one or more transmit antennas. Firstly, we demonstrate that a non-coherent ambiguity function (AF) allows significant improvement in the localization performance compared to the coherent AF previously used for this purpose, especially at high frequencies typically used in communication systems. Secondly, we propose a two-step (coarse-to-fine) localization technique. The second step provides a refined spatial sampling of the AF in the vicinity of its maximum found on the coarse space grid covering an area of interest (in range and depth), computed at the first step. This technique allows high localization accuracy and reduction in complexity and memory storage, compared to single step localization. Thirdly, we propose a joint refinement of the AF around several maxima to reduce outliers. Numerical experiments are run for validation of the proposed techniques.

## 1. Introduction

In recent years, underwater acoustic (UWA) communication in sensor networks has attracted significant interest due to a wide range of commercial and military applications [1,2,3,4,5,6,7,8]. Typical applications are search and rescue [9], environmental and biological monitoring [10], sea floor mapping [11], mining exploration [12] and oil and gas exploration [13]. Underwater localization for acoustic sensors is considered as a major task for those applications since the information collected by the sensors is often associated with the node location. Thus underwater sensors, in particular, sensors with communication transceivers within sensor nodes, require accurate localization. Another important application where the information about the receiver location is essential is the transmit beamforming (also called the antenna precoding) in multiuser underwater acoustic communication networks [14]. Such beamforming can significantly increase the network throughput, and the use of the receiver location allows achieving this goal without a high data-rate feedback communication channel.

However, underwater localization is a challenging task since the techniques used in terrestrial radio systems, such as the global positioning system (GPS), cannot be operated underwater due to the strong attenuation of radio waves [15,16,17]. For underwater localization, matched field processing (MFP) is an effective technique and has been widely investigated [18,19,20,21]. MFP exploits an acoustic model to calculate the field replica, from an acoustic source, to match the field measured by an array of hydrophones [22,23]. A measure of the match defines an ambiguity function (AF) computed on a grid of points in space (range and depth) covering the area of interest. The peak of the AF indicates the estimate of the source location [20,24]. In MFP, the coherent AF is most often used and it is effective at low frequencies, e.g., up to 1 kHz [19,21]. However, the MFP with a coherent AF may lead to false localization estimates (outliers) at high frequencies, at which UWA communications systems typically operate. In [19], it is suggested that a non-coherent AF is used for MFP localization of a UWA communication receiver. Using numerical and real experiments, it is shown in [19] that, with a non-coherent AF, the localization accuracy at high frequencies (8–16 kHz) significantly improves.

The work in [14] considers the underwater localization in a communication network with multiple transmit antennas at the base station and single-antenna receivers at the network nodes. The purpose of the localization is to reduce the amount of data representing the channel state information sent from nodes back to the base station for the transmit antenna precoding. In this work, the coherent AF is used for the MFP localization, and therefore, a large number of transmit antennas and dense spatial sampling are required, resulting in high complexity and high memory storage requirements.

In this paper, we propose and investigate MFP techniques for localization of a single-antenna UWA communication receiver relative to one or more transmit antennas in a number of scenarios. The contributions of this paper are as follows.
We demonstrate that a non-coherent AF allows significant improvement in the localization performance compared to the coherent AF previously used for this purpose, especially at high frequencies.A two-step (coarse and fine steps) technique is proposed. The first step is to find the AF maximum by comparing the estimated channel frequency response with the pre-computed frequency responses in the grid map; the second step provides a refined spatial sampling of the AF in the vicinity of its maximum found on the coarse space grid covering an area of interest (in range and depth), computed at the first step. This technique allows high localization accuracy and a reduction in complexity and memory storage, compared to single step localization.A joint refinement of the AF in the vicinities of several maxima is proposed to reduce outliers.For validation of the proposed techniques, we run numerical experiments in different UWA environments, with different parameters of spatial sampling, number of transmit antennas and different accuracy for the estimation of the acoustic channel response.

This paper is organized as follows. Section 2 introduces the background for receiver localization based on the work in [14]. Section 3 presents the non-coherent metric and describes the refinement approach. Simulation results are presented in Section 4. Section 5 ends with discussion and concluding remarks.

The following notation is used. Boldface upper case letters denote matrices, boldface lower case letters denote column vectors and standard lower case letters denote scalars. The superscript ${\left(\;\cdot \;\right)}^{H}$ denotes the Hermitian transpose, ${∥\;\cdot \;∥}_{2}$ is the Euclidean norm and ⊙ is the Hadamard product.

## 2. Background

In this section, we present a communication scenario and MFP localization technique exploiting a coherent AF.

Consider an UWA environment, where a geographical area of interest is defined as illustrated in Figure 1. We assume that the UWA environment is perfectly known, in particular, the sound speed profile (SSP) is available. The area of interest is covered by grid points, each at a specific sea depth and range from the transmit antenna. Using an acoustic model and the UWA environment parameters, the channel response between the transmit antenna and every grid point is computed and stored in memory. We will call the result of this computation a grid map for the transmit antenna. Such computations are repeated for every transmit antenna and the corresponding grid maps represent a dictionary. This dictionary is available at the receiver, which is located within the area of interest. Using the signal transmitted from each transmit antenna, the receiver estimates the channel responses and compares them with the channel responses of the corresponding grid maps. The best match is assumed to indicate the grid point closest to the true receiver location.

In [14], it is assumed that by using a feedback communication channel, the node sends the grid point index to the base station, where the dictionary channel responses are used for optimization of transmit antenna beamforming. However, this information can also be used for other applications, e.g., attributing the information from sensors to geographical locations. In this paper, we only consider the localization problem.

For comparison of channel estimates with entries in the dictionary, different metrics can be used. For communication systems, it is typical to describe the channel as a linear filter with an impulse response or corresponding frequency response. Let $g_{m}$ be a K×1 vector representing the channel frequency response, corresponding to the *m*th grid point. Elements of the vector are *K* samples of the frequency response (*K* subcarrier amplitudes) within the frequency bandwidth of the communication system. Let $\hat{h}$ be a K×1 estimate of the channel frequency response at the receiver. For comparison of these two vectors, the following metric can be used [14]:(1)$$c_{m}=\frac{|g_{m}^{H}\hat{h}{|}^{2}}{∥g_{m}{∥}_{2}^{2}{∥\hat{h}∥}_{2}^{2}},\;m=1,\cdots ,M,$$
where *M* is the number of grid points in the grid map. The set of values $c_{m}$ over *m* represents a coherent AF. The best match between the channel response estimate and channel responses in the grid map is given by
(2)$$m_{o}=\arg\max_{m=1,\ldots ,M}c_{m},$$
where the grid point index $m_{o}$ defines the receiver location estimate.

With the knowledge of the specific acoustic environment including the SSP, acoustic parameters of the sea bottom, the depth of transmit antennas, and the position of the grid point, a ray tracing acoustic simulator is used to compute $g_{m}$. Elements of the vector $g_{m}$ are then given by
(3)$$g_{m}\left(f_{k}\right)=\sum_{i=0}^{L_{m}-1}A_{m,i}e^{-j2\pi f_{k}\tau_{m,i}},$$
where $f_{k}$, k=0,…,K−1, are subcarrier frequencies at which the channel frequency responses are computed, $L_{m}$ represents the number of rays, $A_{m,i}$ is the complex-valued amplitude and $\tau_{m,i}$ is the delay of the *i*th ray on the *m*th grid point. For our simulation below, the ray information is generated by the BELLHOP3D ray tracing program [25].

However, there is an unknown propagation delay τ between the channel response estimate and channel responses in the grid map. This delay is due to the fact that the pilot transmission and reception are not synchronized and there is an unknown delay between the channel impulse response estimate at the receiver and the impulse responses pre-computed on the grid map using the wave equation [14]. Therefore, in practice, we can only compare shapes of the pre-computed impulse responses and the channel response estimate. Thus, we need to find the best delay shift between these two, for which the covariance is maximized, and this maximum covariance is the measure of similarity of the impulse responses.

In the frequency domain, at a frequency *f*, according to the time-shifting theorem [26], this delay is represented as a factor $e^{-j2\pi f\tau }$. With the unknown delay τ, the measure for comparison of channel frequency responses is given by
(4)$$c_{m}=\frac{\max_{\tau \in [\tau_{\min},\tau_{\max}]}{\left|g_{m}^{H}\Lambda_{\tau }\hat{h}\right|}^{2}}{∥g_{m}{∥}_{2}^{2}{∥\hat{h}∥}_{2}^{2}},$$
where
$$\Lambda_{\tau }=\left[\begin{array}{ccc} e^{-j2\pi f_{0}\tau } & & 0\\ & \ddots & \\ 0 & & e^{-j2\pi f_{K-1}\tau } \end{array}\right],$$

$\Lambda_{\tau }$ is a K×K diagonal matrix. The parameters $\tau_{\min}$ and $\tau_{\max}$ define the delay uncertainty interval. The metric (Equation 4) describes a coherent AF $\left\{c_{m}\right\}$, which provides an improved location estimate $m_{o}$ compared to the AF in (Equation 1).

Computation in (Equation 4) can be efficiently done using the fast Fourier transform (FFT),
(5)$$c_{m}=\frac{\max_{i=1,\ldots ,pK}{\left|q(i)\right|}^{2}}{∥g_{m}{∥}_{2}^{2}{∥\hat{h}∥}_{2}^{2}},$$
where q(i) are elements of the vector q=Fη and η is obtained by zero-padding the vector $g_{m}^{H}⊙\hat{h}$, F is a pK×pK discrete Fourier transform (DFT) matrix and *p* is an integer, p≥1. Using p>1 allows improvement in the delay resolution.

Note that the AF for a particular transmit antenna can have multiple maximums close in magnitude. Since the channel estimates are corrupted by noise, a wrong (local) AF maximum can be chosen for the localization, resulting in outliers. The probability that AFs computed for different antennas have the same positions of maximums is low, which can be exploited to reduce the outliers.

Therefore, with multiple transmit antennas, the localization performance could be further improved by using the AF
(6)$$c_{m}=\sum_{t=1}^{N_{T}}\frac{\max_{\tau \in [\tau_{\min},\tau_{\max}]}{\left|g_{t,m}^{H}\Lambda_{\tau }{\hat{h}}_{t}\right|}^{2}}{∥g_{t,m}{∥}_{2}^{2}{∥{\hat{h}}_{t}∥}_{2}^{2}},$$
where $N_{T}$ is the number of transmit antennas, $g_{t,m}$ is the channel frequency response vector at the *m*th grid point on the *t*th grid map and ${\hat{h}}_{t}$ is the estimate of the channel frequency response between the *t*th transmit antenna and receiver antenna.

Figure 2 shows the coherent AF defined in (Equation 6) for an acoustic environment described in Table 1. Specifically, the SSP is uniform (sound speed is consistent, 1500m/s) as shown in Figure 3, the number of transmit antennas $N_{T}=4$, the area of interest in range is from 100 m to 220 m and in depth from 30 m to 100 m, the grid steps in both range and depth are 1 m. It can be seen in Figure 2 that the true position of the receiver is at the range of 184.5 m and in 70 m depth. However, the maximum of the AF is found at the depth 41 m and range 108 m. It can be seen that the location estimate is very poor, the estimate is about 82 m away from the true location. This happens because the spatial sampling interval is too large to provide accurate representation of the AF, i.e., the AF samples miss the AF maximum. To overcome this problem, we need to reduce the spatial sampling interval, so that we do not miss the AF maximum. However, this results in a higher number of grid points *M*, and thus the memory required for saving the dictionary increases and the complexity of the AF computation in (Equation 6) also increases. In order to keep the memory and complexity low, a non-coherent AF is proposed as described in Section 3.

## 3. Non-Coherent AF and Refinement

In this section, we introduce a non-coherent AF and demonstrate its efficiency for the localization in comparison to the coherent AF and describe the proposed coarse-to-fine localization approach.

### 3.1. Non-Coherent AF

The coherent AF requires dense spatial sampling, which results in a high computation complexity and large memory storage for saving the dictionary. The example of the coherent AF in Figure 2 shows that even with a relatively low carrier frequency $f_{c}=3072\;\mathrm{Hz}$ and small grid step $C_{d}=C_{r}=1\;m$, false localization (outlier) can happen when the receiver is located between grid points.

A better localization performance can be achieved with the non-coherent AF defined as
(7)$$c_{m}=\sum_{t=1}^{N_{T}}\frac{\max_{\tau \in [\tau_{\min},\tau_{\max}]}|{\tilde{g}}_{t,m}^{H}\Lambda_{\tau }{\tilde{h}}_{t}{|}^{2}}{||{\tilde{g}}_{t,m}|{|}_{2}^{2}||{\tilde{h}}_{t}|{|}_{2}^{2}},$$
where ${\tilde{g}}_{t,m}=\tilde{F}\mathrm{abs}\left({\tilde{F}}^{H}g_{t,m}\right)$, ${\tilde{h}}_{t}=\tilde{F}\mathrm{abs}\left({\tilde{F}}^{H}h_{t}\right)$, $\tilde{F}$ is the K×K DFT matrix and the vector function abs(g) is defined as
$$\mathrm{abs}\left(g\right)=\left[\begin{array}{c} |g_{1}|\\ \dot{.}\\ |g_{K}| \end{array}\right],$$
where $g_{K}$, k=1,…,K, are elements of the vector g.

This AF is based on comparison of magnitudes of channel impulse responses, and thus the phase information is removed from the comparison.

Figure 4 shows an example of receiver localization using the non-coherent AF in (Equation 7) for the parameters of acoustic environment in Table 1. When comparing Figure 2 and Figure 4, it can be seen that the non-coherent AF is significantly smoother than the coherent AF and the maximum of the non-coherent AF provides an accurate estimate of the receiver location.

### 3.2. Refinement

The receiver location can be estimated on the grid map using the coarse localization scheme. However, the accuracy of the coarse estimation is limited by the coarse grid steps; additionally, outliers are more likely when the receiver is not located on a grid point. Therefore, a fine estimation of the receiver location is required to reduce the error between the estimated and true receiver positions. For the refinement, the estimated position resulted from the coarse estimation is regarded as a center point, and a small-size (refined) grid map around the center point is generated with a finer resolution.

The localization performance can be improved by the refinement of the grid map in the vicinity of the coarse estimate. Figure 5 demonstrates how the refinement works. The sign △ indicates the true receiver position. The sign ■ indicates the maximum of the AF on the coarse grid. Assuming that this is not an outlier, these two positions will be close to each other as shown in Figure 5. For the refinement, additional grid points are computed with a finer resolution; in Figure 5a, the refined steps in both range and depth are half that of the coarse grid steps. In Figure 5a, the refinement area is chosen as $2C_{r}\times 2C_{d}$. In some cases, as will be shown in Section 4, a larger refinement area can improve the localization performance, e.g., as shown in Figure 5b, where the refinement area is $4C_{r}\times 4C_{d}$. The error of the coarse localization is the distance between the signs △ and ■, whereas the error of the fine localization is the distance between the sign △ and the closest refined grid point, which is smaller than the coarse error due to the use of a small refined step.

The refined grid map can be computed by using the ray tracing model in the same way as the computation of the coarse grid map. However, a computationally more efficient approach is based on the bilinear interpolation between coarse grid points.

Consider an example of the bilinear interpolation of the acoustic field at the refined grid point (*x*, *y*) using the acoustic fields computed at the four neighboring coarse grid points. To compute amplitudes and delays for rays arriving at the point (*x*, *y*), we will be using the approach in [28]. The approach in [28] is illustrated in Figure 6. The vector of amplitudes is given by
$$\begin{array}{c} a=\left[\begin{array}{c} \left(1-w_{1}\right)\left(1-w_{2}\right)a_{1}\\ \left(1-w_{1}\right)w_{2}a_{2}\\ w_{1}w_{2}a_{3}\\ w_{1}\left(1-w_{2}\right)a_{4} \end{array}\right], \end{array}$$
where $a_{j}$ is the $\ell_{j}\times 1$ vector of arrival amplitudes at the *j*th coarse grid point, j=1,…,4. $\ell_{j}⩽\ell_{\max}$, $\ell_{\max}$ defines the maximum number of arrivals. The weights are given by
(8)$$\begin{array}{cc} w_{1}= & \left(x-x_{1}\right)/\left(x_{2}-x_{1}\right),\\ w_{2}= & \left(y-y_{1}\right)/\left(y_{2}-y_{1}\right), \end{array}$$
where $w_{1}$ and $w_{2}$ represent proportional distance in the *x* direction and *y* direction, respectively. The vector of delays is given by
(9)$$d=\left[\begin{array}{c} d_{1}+\Delta d_{1}\\ d_{2}+\Delta d_{2}\\ d_{3}+\Delta d_{3}\\ d_{4}+\Delta d_{4} \end{array}\right],$$
where $d_{j}$ is the $\ell_{j}\times 1$ vector of arrival delays at the *j*th coarse grid point, The adjusted delays from position $\left(x_{j},y_{j}\right)$ to position (x,y) are computed as
(10)$$\Delta d_{j}=\left(\Delta x_{j}\cos\theta_{j}+\Delta y_{j}\sin\theta_{j}\right)/c_{j},$$
where
(11)$$\begin{array}{c} \Delta x_{j}=x-x_{j},\\ \Delta y_{j}=y-y_{j},\; \end{array}$$

$\theta_{j}$ is the $\ell_{j}\times 1$ vector of arrival angles at the *j*th coarse grid point, j=1,…,4, and $c_{j}$ is the sound speed at the depth of the *j*th coarse grid point.

Elements of the frequency response for the *n*th refined grid point, the point (x,y) as shown in Figure 6, are given by
(12)$$g_{n}^{m_{o}}\left(f_{k}\right)=\sum_{i=0}^{\ell_{1}+\ell_{2}+\ell_{3}+\ell_{4}-1}a_{i}e^{-j2\pi f_{k}d_{i}},$$
where k=0,⋯,K−1, $a_{i}$ and $d_{i}$ are elements of vectors a and d, respectively. The vector $g_{n}^{m_{o}}$ with elements from (Equation 12) is used to compute the AF $c_{n}^{m_{o}}$.

With such a refinement, an improved position estimate is found from the maximum AF within the refinement area:(13)$$n_{o}=\arg\max_{n=1,\cdots ,M_{R}}c_{n}^{m_{o}},$$
where $M_{R}$ is the number of refined grid points in the vicinity of the coarse receiver location estimate $m_{o}$ and the set of values $c_{n}^{m_{o}}$ over *n* from 1 to $M_{R}$ is the AF computed on the refined grid map. For the refined area in Figure 5a, $M_{R}=25$; for the refined area in Figure 5b, $M_{R}=81$. As will be shown in Section 4.2, the refinement can greatly reduce the error between the estimated and true receiver positions.

### 3.3. Multiple Refinement Areas

The receiver position is found as the position of the global AF maximum. The AF, as a continuous function of range and depth, apart from the global maximum, has multiple local maxima. With a finite spatial sampling rate, i.e., finite grid steps in range and depth, the AF maximum on the grid map might correspond to a local maxima. In this situation, the location estimate is an outlier, i.e., the location error can be arbitrary high. The refinement does not overcome this problem since it is possible the refinement is performed in the vicinity of the outlier.

In order to solve this problem, we can choose several AF maxima, the number of which is defined as $N_{\max}$, from the coarse grid map, perform refinement in the vicinity of each of them and find the AF maximum jointly on all the $N_{\max}$ refinement areas.

This can be implemented as illustrated in Figure 7. Firstly, the AF maximum is found on the coarse grid map, the maximum position is $m_{o}^{\left(1\right)}$. Then, coarse grid points in the corresponding refinement area, around the coarse grid point $m_{o}^{\left(1\right)}$, are removed from the coarse grid map. We will consider two cases of removing the coarse grid points. In the first case, only the maximum point is removed (one point). In the second case, 9 points are removed including the maximum and eight neighboring coarse points. Then the AF maximum at the grid position $m_{o}^{\left(2\right)}$ is found on the updated coarse grid map. The same procedure can be repeated to find the third AF maximum at the position $m_{o}^{\left(3\right)}$, etc. For each new grid position with AF maximum, the refinement is now performed in the vicinity of the possible candidate for receiver location. The position of a joint AF maximum over $N_{\max}$ multiple refinement areas is the final location estimate. As will be shown in Section 4.3, the multiple refinement can remove outliers in the localization process.

### 3.4. Complexity of the Two-Step Localization

In this subsection, we present an analysis of the complexity of the proposed localization technique.

For the localization, the following steps should be done:

1. Coarse step, including the AF computation in (Equation 7) processed by an efficient algorithm in (Equation 5), and finding the AF maximum.

2. Refinement step, including the computation of $N_{\max}$ refined grid maps with the bilinear interpolation described by (Equation 9) to (Equation 12), computation of the refined AFs and their maxima using (Equation 5), (Equation 7) and (Equation 13).

Specifically, for the coarse step, in (Equation 7), the vector ${\tilde{g}}_{t,m}$, related to the channel frequency response on the *m*th grid point corresponding to the *t*th coarse grid map, can be pre-computed and stored into memory. Here, we consider the computation of the vector ${\tilde{h}}_{t}$, related to the estimated channel frequency response. ${\tilde{h}}_{t}=\tilde{F}\mathrm{abs}\left({\tilde{F}}^{H}h_{t}\right)$ requires two FFT operations of size *K*; when using the split-radix FFT algorithm in [29], the complexity of computing each FFT requires $K\log_{2}K$ multiply and accumulate operations (MACs). The complexity of computing $\mathrm{abs}\left({\tilde{F}}^{H}h_{t}\right)$, requires 6K MACs. In (Equation 5), the computation is considered for every grid point in each grid map. The complexity of computing $\tilde{q}$ requires the FFT operation of size pK, which requires $pK\log_{2}pK$ MACs; the complexity of computing ${\tilde{g}}_{m}^{H}⊙\tilde{h}$ is *K* MACs; the complexity of computing square of elements in $\tilde{q}$, $|\tilde{q}{|}^{2}$, requires 2pK MACs; the complexity of computing the maximum requires pK MACs; for the $∥\tilde{h}{∥}_{2}^{2}$, the complexity of this computation is about *K* MACs. Therefore, the complexity of computing the coarse receiver localization for each trial is
(14)$$C_{\mathrm{coarse}}\approx N_{T}\left[2K\log_{2}K+6K+M\left(pK\log_{2}pK+3pK+2K\right)\right].$$

For the refined step, based on the *t*th coarse grid map, the complexity of computing (Equation 12) requires $4K\ell_{\max}$ for every refined point corresponding to each local maxima. The complexity of computing the refined AFs using (Equation 7) and (Equation 5) is the same as the coarse step for each point, it requires $2K\log_{2}K+6K+pK\log_{2}pK+3pK+2K$ MACs. Therefore, the complexity of computing the fine receiver localization for each trial is given as,
(15)$$C_{\mathrm{fine}}\approx N_{T}N_{\max}M_{R}\left(4K\ell_{\max}+8K+2K\log_{2}K+pK\log_{2}pK+3pK\right).$$

The total complexity of computing the coarse-to-fine receiver localization is
(16)$$C_{\mathrm{total}}=C_{\mathrm{coarse}}+C_{\mathrm{fine}}.$$

The complexity for coarse-search computation and fine-search computation is shown in Figure 8. Figure 8a shows the complexity of the coarse localization algorithm with different number of transmit antennas $N_{T}$. For the whole area of interest with $M=201\times 501\approx 10^{5}$ coarse grid points, the complexity of the coarse search for $N_{T}=4$ transmit antennas, $C_{\mathrm{coarse}}$ $\approx 5.7\times 10^{10}$ MACs. This complexity may be excessive for a general-purpose processor, especially the ones that can be practically used on low-power communication nodes. However, most of the computation is based on the FFT and vector multiplication, i.e., operations well suited to implementation as hardware accelerators [30,31]; moreover, since the coarse search involves multiple parallel computations, its hardware implementation, e.g., on Field Programmable Gate Array (FPGA) design platforms can be very efficient, making this stage of the proposed localization algorithm feasible. As for the coherent AF, as was mentioned in [14], the number of the grid points *M* needs to be significantly higher even for such a low carrier frequency as $f_{c}=3072\;\mathrm{Hz}$, thus making the coarse search less suitable for practical implementation than that with the non-coherent AF.

A reduction in the coarse-search computation can be achieved by using a pre-localization of the receiver by any known methods. e.g., the knowledge of the receiver depth can significantly reduce the grid size *M* and, as will be shown in Section 4.4, also results in a higher localization accuracy. As an example, from Figure 8a, it is seen that with four coarse grid points in depth, the total number of grid points is reduced to M=4×501≈2000; in this case, $C_{\mathrm{coarse}}\approx 1.1\times 10^{9}$ MACs, which is more affordable at the receiver node.

Figure 8b shows the complexity of the fine localization algorithm with different combinations of the product $N_{T}N_{\max}$. For the highest accuracy, when the refined steps are set to $F_{r}=F_{d}=0.1\;m$, we have $M_{R}=441$ points in a refined area, and with $N_{T}=4$ and $N_{\max}=4$, the fine-search complexity, $C_{\mathrm{fine}}\approx 1\times 10^{9}$ MACs, which is high, but still lower than the complexity of the coarse search. For the lower localization accuracy, when the refined steps are set to $F_{r}=F_{d}=0.5\;m$ ($M_{R}=25$), the fine-search complexity, $C_{\mathrm{fine}}\approx 9\times 10^{6}$ MACs for $N_{T}=1$ and $N_{\max}=1$, which is significantly lower than the coarse-search complexity. Thus, the refinement stage does not result in a significant increase in the total algorithm complexity compared to the coarse-search complexity.

## 4. Numerical Results

In this section, we present results of numerical experiments. The objectives of the numerical experiments are:Comparison of the coarse localization accuracy using the coherent and non-coherent AFs.Analysis of the coarse-to-fine localization performance.Analysis of using multiple refinement areas for the localization.Analysis of robustness of the localization to the mismatch between the acoustic environment used for computation of the dictionary and true acoustic environment.Analysis of robustness of the localization to the channel estimation errors due to the noise.

In the experiments, to measure the localization performance, the cumulative distribution function (CDF) is computed for the position error
(17)$$\varepsilon =\sqrt{{\left(\hat{x}-x\right)}^{2}+{\left(\hat{y}-y\right)}^{2}},$$
where $\hat{x}$ and $\hat{y}$ are estimates of the true range *x* and depth *y*, respectively. The CDF is obtained in 100 simulation trials. In each simulation trial, the receiver position is uniformly random within the area of interest. The main simulation parameters are given in Table 2.

### 4.1. Coarse Localization Using Coherent and Non-Coherent AFs

In this subsection, we compare the coarse localization performance using the coherent AF and non-coherent AF metrics. Figure 9 shows the localization performance of using coherent and non-coherent AF at two carrier frequencies with number of transmit antennas $N_{T}$ varying from 1 to 4. It can be seen that the localization performance provided by the non-coherent AF is significantly better than that provided by the coherent AF. With the increase of the number of transmit antennas $N_{T}$, the performance improves for both metrics. At the low carrier frequency $f_{c}=3072\;\mathrm{Hz}$, when using the non-coherent AF with $N_{T}=2$, all receivers are localized within an error ε≤2 m, whereas, for the coherent AF even with $N_{T}=4$, in more than 40% of cases, the error is higher than 2 m. Thus, the use of the non-coherent AF significantly reduced the number of outliers, as was previously demonstrated in Figure 4. It can also be seen in Figure 9 that the increase of the carrier frequency $f_{c}$ results in significant degradation of the localization performance with the coherent AF, whereas, for the non-coherent AF, the localization performance is consistent.

### 4.2. Coarse-to-Fine Localization

We now demonstrate the benefit of the refinement for improving the localization performance. Figure 10 shows the localization accuracy against the refinement steps in both depth and range with two sizes of refinement areas. It can be seen that the localization accuracy proportionally improves with the reduction in the refinement step, as long as there are no outliers. Figure 10a presents results for one transmit antenna, and the refinement area in Figure 5a. It is seen that if the non-coherent AF maximum on the coarse grid map is found in one of four coarse grid points surrounding the grid cell where the receiver is positioned, i.e., ε<1.4m, then the localization accuracy improves proportionally to the reduction in the refined step. It is also seen that there is a “step” in the CDF at ε≈1.4m. This error corresponds to the maximum distance within the grid cell, and this means that, in a significant number of the trials, the non-coherent AF maximum is found in neighbouring grid cells.

Figure 11 illustrates one such case. The increase in the refinement area from 2m×2m to 4m×4m (as shown in Figure 5b), improves the localization accuracy as can be seen from the comparison of Figure 10a,b. The increase in the refinement area allows somewhat reduction in “small” outliers. The number of outliers can also be reduced by increasing the number of transmit antennas, as demonstrated in Figure 10c. In this case, not only “small”, but also “large” outliers are also eliminated. It will be shown in Section 4.3 that the probability of outliers can be significantly reduced when using multiple refinement areas.

### 4.3. Multiple Refinement Areas

We now demonstrate the benefit of using multiple refinement areas for improving the localization performance, primarily by reducing the probability of outliers. Figure 12 shows the localization accuracy against the number of refinement areas $N_{\max}$. In this experiment, after finding an AF maximum on the coarse grid map, the maximum point is removed before the search for the next maximum. It can be seen that even with such large range and depth refined steps ($F_{r}=F_{d}=0.5\;m$), the search in two refinement areas significantly reduces the probability of outliers. Further increase in the number of refinement areas to $N_{\max}=4$ provides further significant improvement; the probability of localization error ε<1m is as high as 96%. Note that this performance is achieved with only one transmit antenna. With two transmit antennas, as can be seen in Figure 13, there is no outliers and the localization error is lower than 0.5m in all simulation trials.

In Figure 14, we compare the localization performance with multiple refinement areas at different refined steps using four transmit antennas. With $N_{T}=4$ and $N_{\max}=4$, the localization accuracy depends only on the refined step size, the smaller the refined step, the higher accuracy can be achieved.

We now present results for another acoustic environment, with the SSP from the SWellEx-96 experiment [32] shown in Figure 3.

Recall that in Figure 12, we showed the localization performance with different number and size of refinement areas for the uniform SSP. The refined step used is 0.5m. Figure 15 shows the localization performance for the SWellEx-96 SSP. As can be seen from comparison of results in Figure 12 and Figure 15, the coarse localization performance with one transmit antenna with the SSP from the SWellEx-96 experiment provides significantly more outliers than that with the uniform SSP environment. For the uniform SSP, 85% of cases have the error ε<1.4m, whereas, for the SWellEx-96 SSP, only 58% of cases have such localization accuracy. By adopting four refinement areas of size 2m×2m, the probability of outliers in the SWellEx-96 SSP environment is reduced from 42% to 26%, while in the uniform SSP environment, it is reduced from 15% to 4%. Even with a larger refinement area of 4m×4m, in the SWellEx-96 SSP environment, the probability of outliers is still as high as 14%.

In Figure 16, we show the localization performance with the SWellEx SSP using four transmit antennas. Two cases are considered as described in Section 3.3. For the first case, only one point is removed from the coarse grid map after finding the AF maximum; in the second case, nine points are removed. It can be seen that the use of four transmit antennas allows significant reduction in the number of outliers. The localization accuracy can be further improved by using a smaller refined step. It also can be seen that when a smaller refined step is used, the localization performance can be further improved by removing more (nine) points from the coarse grid map before finding the next maximum. This can be explained by the fact that positions of several maxima are close to each other resulting in overlapping refinement areas, thus reducing the probability of finding the global maximum. For the rest of the paper, we adopt the case of removing nine points from the coarse grid map when a smaller refined step (0.1 m) is used.

### 4.4. Mismatched Environments

In this subsection, we consider scenarios with mismatched environments when acoustic parameters used for computation of the dictionary differ from real acoustic parameters.

In the first experiment, the dictionary is computed using the SWellEx-96 SSP, while the true SSP used in the experiment is given by
(18)$$\mathrm{SSP}\left(i\right)=\mathrm{SSP}\left(i\right)+n\left(i\right),i=1,\cdots ,N_{d},$$
where $N_{d}$ is the number of depth points with SSP values, *i* is the index of the corresponding depth, n(i) are independent Gaussian random numbers with a variance of $\sigma_{\mathrm{ssp}}^{2}$.

The SWellEx-96 SSP and a realization of the mismatched SSP used in the experiment in the case $\sigma_{\mathrm{ssp}}=1\;m/s$ is shown in Figure 3. Figure 17 shows the localization performance for different levels of the SSP mismatch. It can be seen that the localization performance is close to the matched performance for $\sigma_{\mathrm{ssp}}\leq 1\;m/s$. The performance degrades for a higher level of mismatch of the SSP ($\sigma_{\mathrm{ssp}}=3\;m/s$). It can be concluded that the localization performance is robust against the small mismatch of the SSP.

To reduce the sensitivity of a mismatched model, we consider a scenario when the depth of the receiver is known. This can be easily achieved in practice by using a receiver equipped with a depth sensor. Figure 18 shows the localization performance of a mismatched acoustic environment with $\sigma_{\mathrm{ssp}}=1\;m/s$, with and without knowledge of the receiver depth. It can be seen in Figure 18a that the localization performance can be improved by increasing the number of transmit antennas to $N_{T}=4$. Further improvement of the performance can be observed by increasing the size of refinement areas to 4m×4m, resulting in the localization error ε to be smaller than 0.3m in 99% of the trials. In Figure 18b, we assume the depth of a receiver is known. It can be seen that the localization performance improves, the most significant improvement is observed when we use $N_{T}=4$ and a small refinement area of 2 m × 2 m.

In Figure 19, we consider the scenario where the dictionary is computed assuming a flat sea surface, whereas the “real” sea surface is a sinusoid of amplitude $A_{\sin}$ and a period of 8 m. We consider a range of sea surface amplitudes from 0.01m to 0.5m. It can be seen that the higher the sea surface amplitude the higher is the localization error. However, for $A_{\sin}<0.2\;m$, in all simulation trials, the localization error is smaller than 2m, which is acceptable for many applications.

### 4.5. Inaccurate Channel Estimation

In this experiment, the channel frequency response $\tilde{h}\left(f\right)$ between a transmit antenna and the receiver at a frequency *f* is given by
(19)$$\tilde{h}\left(f\right)=\hat{h}\left(f\right)+n\left(f\right).$$

The estimated channel frequency vector $\tilde{h}$ is now represented as $\tilde{h}={\left[\tilde{h}\left(f_{0}\right),\cdots ,\tilde{h}\left(f_{K-1}\right)\right]}^{T}$. The noise samples $n(f_{k})$ are independent complex-valued random Gaussian numbers with zero mean and variance $\sigma^{2}$. The signal-to-noise ratio (SNR) of the channel response estimate is defined as
(20)$$\mathrm{SNR}=\frac{1}{\sigma^{2}}\frac{1}{K}\sum_{k=0}^{K-1}{\left|\hat{h}\left(f_{k}\right)\right|}^{2}.$$

Figure 20 shows localization results against the SNR of the channel response estimate. It can be seen that for SNR=10 dB, the localization results are close to that of the perfect channel response estimation. However, even for the SNR as low as SNR=5 dB, in all the trials the localization error is smaller than 2 m. This demonstrates the robustness of the localizations performance against the estimation error of the channel response.

## 5. Discussion and Conclusions

In this paper, we continue investigation of localization of a receiver relative to transmitter in a communication system using the MFP approach. We have proposed new localization techniques for a single-antenna UWA communication receiver. Specifically, a non-coherent AF has been proposed to improve the localization accuracy, especially at high frequencies. Furthermore, a two-step (coarse-to-fine) localization technique has been proposed. A joint refinement scheme with multiple refinement areas has also been proposed to reduce the number of outliers and to improve the localization accuracy. The performance of the proposed techniques has been evaluated in numerical simulations. The robustness of the localization performance has also been investigated when there is a mismatch of the acoustic environment or under different levels of channel estimation accuracy.

The MFP in a communication system benefits from the knowledge of the transmitted (pilot) signal, compared to its application in sonar systems where normally the source signal is unknown, and thus can potentially provide a higher localization accuracy. Another difference is that the MFP in sonar systems is based on multiple receive antennas, whereas in communication systems it can be used with single/multiple transmit and single/multiple receive antennas. In this paper, we focused on scenarios with multiple transmit and a single receive antennas, whereas the case of a single transmit antenna is of a special interest since it is applicable in most communication systems. Our simulation results suggest that even in the case of the single transmit antenna, when using the non-coherent AF, it is possible to achieve useful results for localization of the single-antenna receiver.

In noisy environments, the coherent AF can potentially provide a better localization performance than the non-coherent AF. However, to realise this benefit, the coherent AF needs to be sampled with a space interval smaller than the wavelength, i.e., the interval is inversely proportional to the carrier frequency of the communication system. The space sampling interval for the non-coherent AF depends on the frequency bandwidth. Since, in a typical communication system, the frequency bandwidth is much smaller than the carrier frequency, the number of grid points covering an area of interest, for the non-coherent AF, is significantly reduced and consequently the amount of computation required for the localization is also significantly reduced, thus making the use of the non-coherent AF more practical. Moreover, the non-coherent AF also results in a smaller memory required for saving the information on the grid.

Although the non-coherent AF can reduce the number of grid points, this number still can be too high for real-time implementation in a communication receiver limited in computation resources. Further reduction in computation can be achieved by using a pre-localization of the receiver by any of known methods. For example, the knowledge of the receiver depth can significantly reduce the grid size and, as was shown in this paper, also results in a higher localization accuracy.

The spatial refinement and multiple refinement proposed in this paper can achieve a very high localization accuracy, thus compensating for possibly low space resolution at the coarse grid when using the non-coherent AF. Since the refinement areas are typically much smaller than the whole localization area, this improvement is achieved with relatively small computations. The joint search over multiple refinement areas allows one to avoid localization outliers that can appear due to errors at the coarse stage in finding the AF area with global maximum.

Most results in this paper have been obtained based on the assumption that the acoustic environment (the sea depth, bathymetry, state of sea surface, SSP, etc.) used for computing the channel state information on the grid is perfectly known. In practice, such knowledge is almost impossible to achieve; note that this is a common problem of the MFP approach. This problem can be partly solved by frequent real-time measurements of the SSP; in particular, this can be done using the MFP inversion techniques [33,34,35,36]. It would be interesting to implement such techniques based on the communication signals, thus reusing the available resources in the communication system. In this paper, we also investigated the loss in the localization accuracy for the cases when: (a) the real SSP differs from that used for the grid computations; (b) the sea surface is not flat; and (c) the channel estimates are distorted by noise. However, more thorough investigation of sensitivity of the localization accuracy to the environment mismatch is still required.

It may happen that the receiver is outside the area covered by the coarse grid; this case has not been addressed in this paper. Another problem is when the acoustic environment is disturbed by the presence of an underwater object. These cases require special consideration. We will continue to deal with these and other problems and would be happy to share our results and data with other research groups.

The ultimate validation of any technique in underwater acoustics can only be done in sea experiments. However, this research topic is still in its infancy and many research problems should be solved, in particular the problems mentioned above, before such experiments can become useful. 

## Figures and Tables

**Figure 1 sensors-22-06968-f001:**
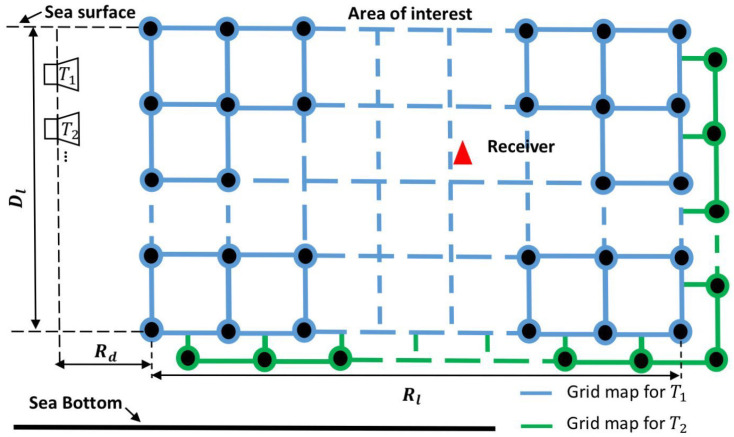
An example of two grid maps for a geographical area; every grid map corresponds to a specific transmit antenna.

**Figure 2 sensors-22-06968-f002:**
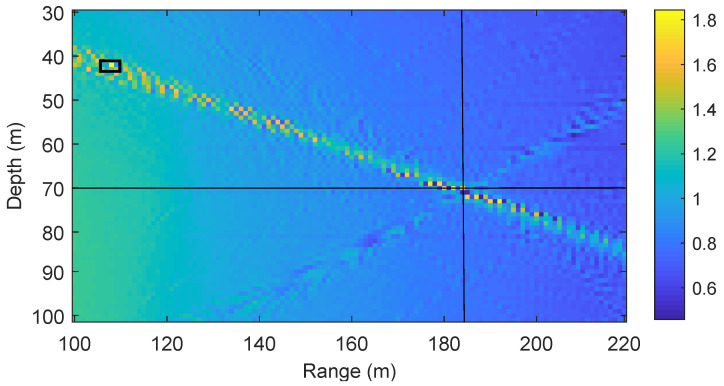
An example of the coherent AF in (Equation 6) for the parameters of acoustic environment in Table 1. The crossing point of the horizontal and vertical black lines indicates the true receiver position. The black square indicates the position estimate (the AF maximum). Here we use the acoustic environment with the uniform SSP as shown in Figure 3.

**Figure 3 sensors-22-06968-f003:**
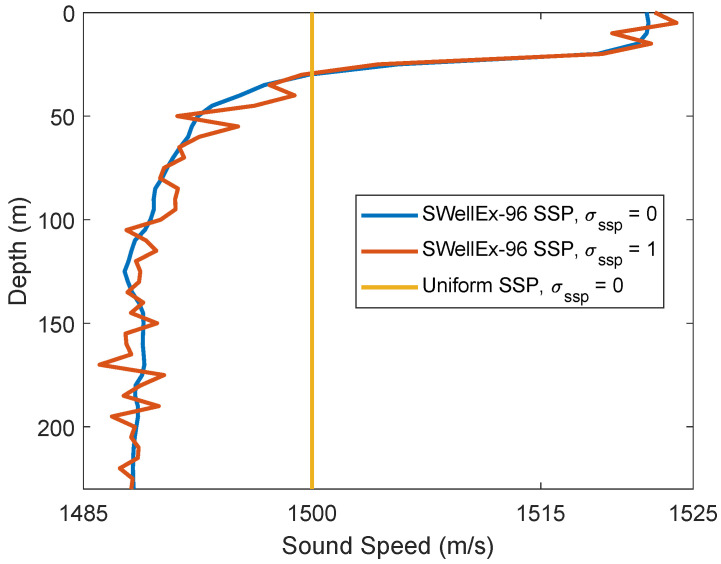
Sound speed profiles (SSPs): uniform, SWellEx-96 [27] and the mismatched SWellEx-96 when the variance sound speed $\sigma_{\mathrm{ssp}}^{2}$ = 1 (m${}^{2}$/s${}^{2}$). (Note: there is no change for the SSP when $\sigma_{\mathrm{ssp}}^{2}=0$, the mismatched is discussed in Section 4.4).

**Figure 4 sensors-22-06968-f004:**
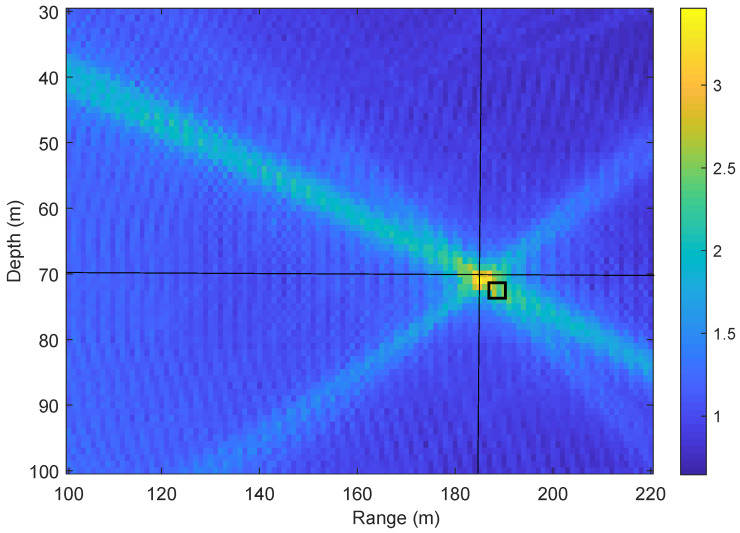
An example of the non-coherent AF in (Equation 7) for the parameters of acoustic environment in Table 1. The crossing point of the horizontal and vertical black lines indicates the true receiver position. The black square indicates the position estimate (the AF maximum). Here we use the acoustic environment with the uniform SSP as shown in Figure 3.

**Figure 5 sensors-22-06968-f005:**
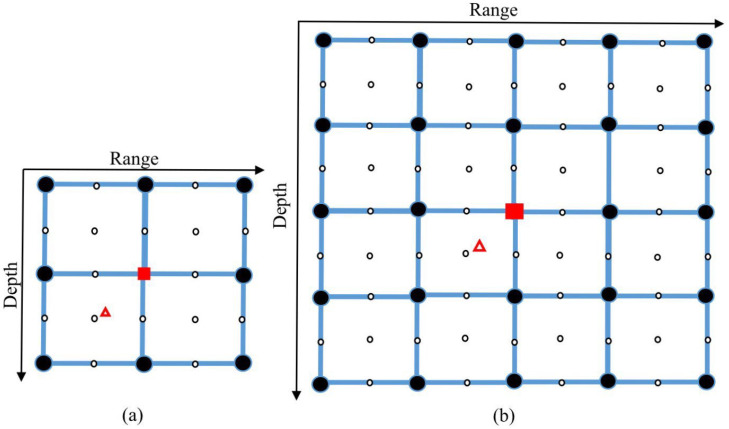
The structure of refined area: (**a**) $2C_{r}\times 2C_{d}$; (**b**) $4C_{r}\;\times \;4C_{d}$. The refined grid step in depth is $F_{d}=C_{d}/2$, the refined grid step in range is $F_{r}=C_{r}/2$. Notation: △ is the true receiver position, ■ is the coarse location estimate, ● are coarse grid points, ○ are refined grid points.

**Figure 6 sensors-22-06968-f006:**
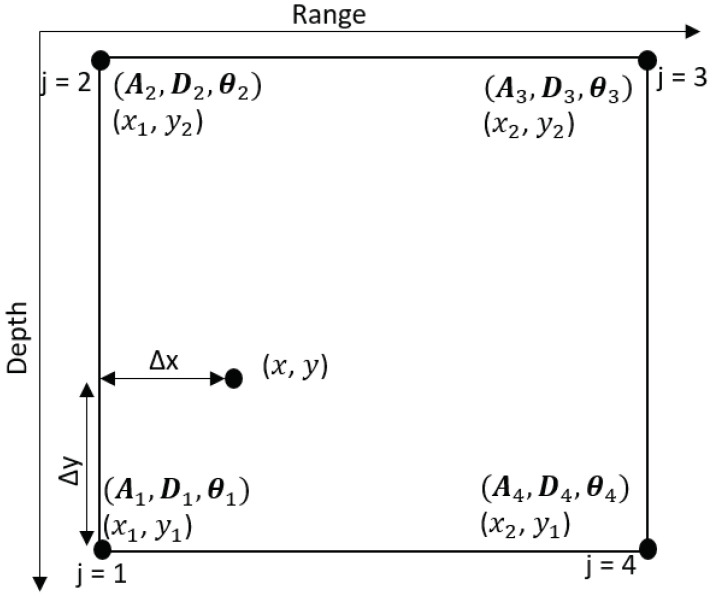
An illustration of the bilinear interpolation. Points $\left(x_{1},y_{1}\right)$, $\left(x_{1},y_{2}\right)$, $\left(x_{2},y_{2}\right)$, $\left(x_{2},y_{1}\right)$ are grid points on the coarse grid map. The point (x,y) is the refined grid point. The vectors $a_{j}$, $d_{j}$, $\theta_{j}$, j=1,…,4, are vectors of the ray amplitudes, delays and angles of arrivals for the *j*th coarse grid point in this figure.

**Figure 7 sensors-22-06968-f007:**
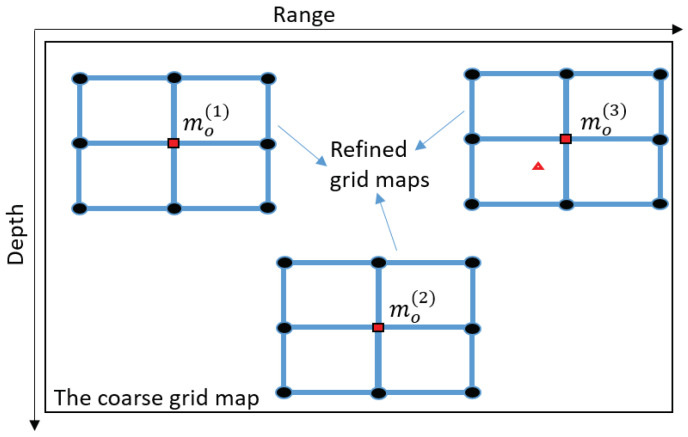
An illustration of multiple refinement areas ($N_{\max}=3$ as an example) in the area of interest. (See notation in Figure 5).

**Figure 8 sensors-22-06968-f008:**
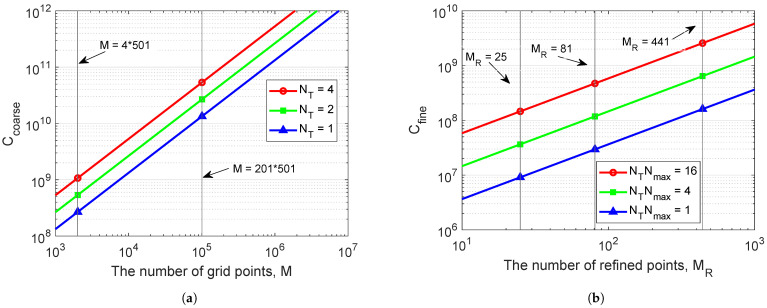
The complexity of the proposed localization algorithm against the number of grid points *M* or refined points $M_{R}$: (**a**) Coarse-search complexity; (**b**) Fine-search complexity.

**Figure 9 sensors-22-06968-f009:**
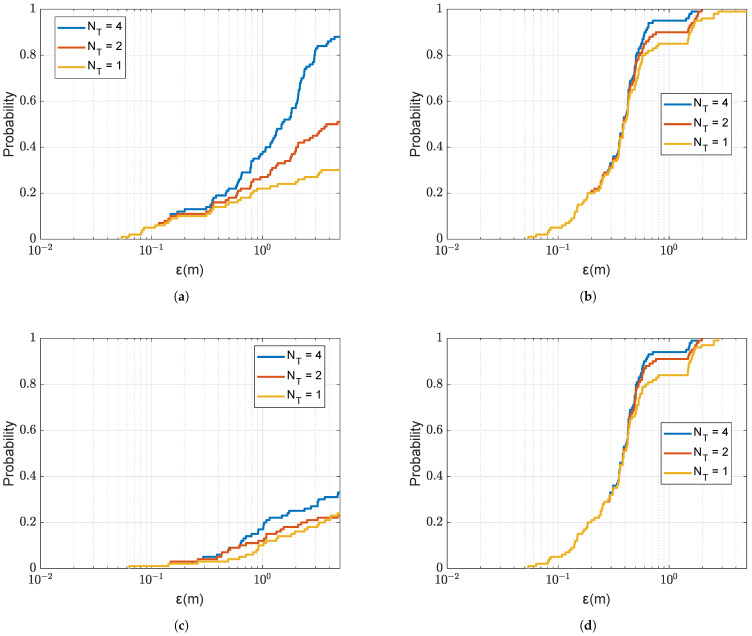
CDF of the localization error ε for the coarse localization using the coherent and non-coherent AFs at low and high carrier frequency $f_{c}$ against the number of transmit antennas $N_{T}$; the SSP is uniform as shown in Figure 3. (**a**) Coherent AF, $f_{c}=3072\;\mathrm{Hz}$. (**b**) Non-coherent AF, $f_{c}=3072\;\mathrm{Hz}$. (**c**) Coherent AF, $f_{c}$ = 15,360 Hz. (**d**) Non-coherent AF, $f_{c}$ = 15,360 Hz.

**Figure 10 sensors-22-06968-f010:**
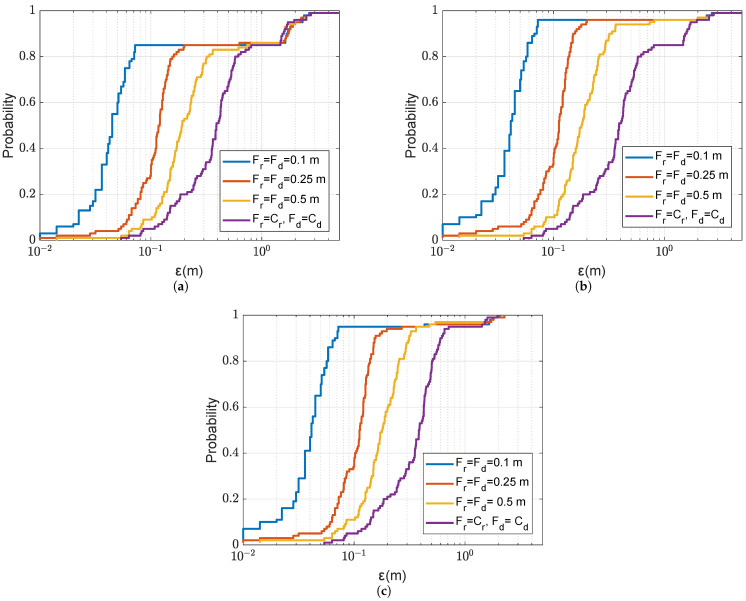
CDF of the localization error ε for the coarse-to-fine localization using the non-coherent AF against different refined steps in range $F_{r}$ and depth $F_{d}$ with two refinement areas of different size as shown in Figure 5; the SSP is uniform as shown in Figure 3. (**a**) $N_{T}=1$, refinement area: 2m×2m. (**b**) $N_{T}=1$, refinement area: 4m×4m. (**c**) $N_{T}=4$, refinement area: 2m×2m.

**Figure 11 sensors-22-06968-f011:**
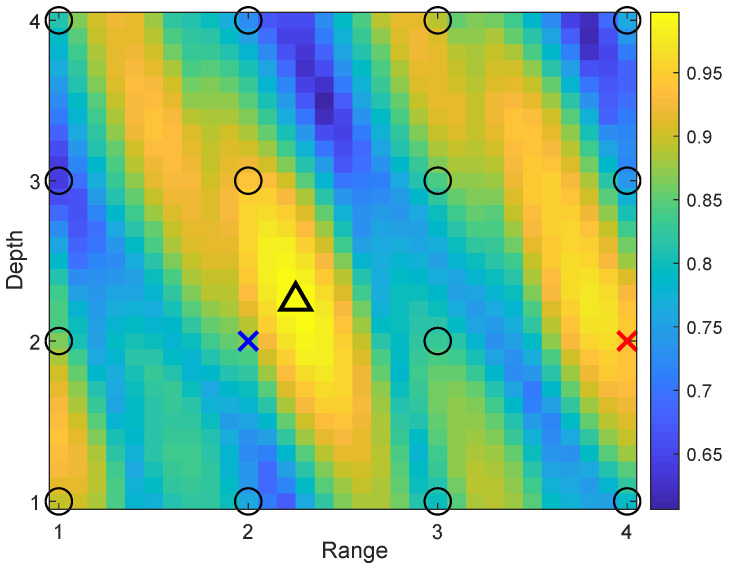
An example of the “continuous-range continuous-depth” non-coherent AF in an area of 3m×3m. The circles are positions of coarse grid points; the triangle is the position of the receiver; the blue cross is the coarse grid point closest to the true position of the receiver; the red cross is the coarse grid point with the AF maximum on the coarse grid map; $N_{T}=1$.

**Figure 12 sensors-22-06968-f012:**
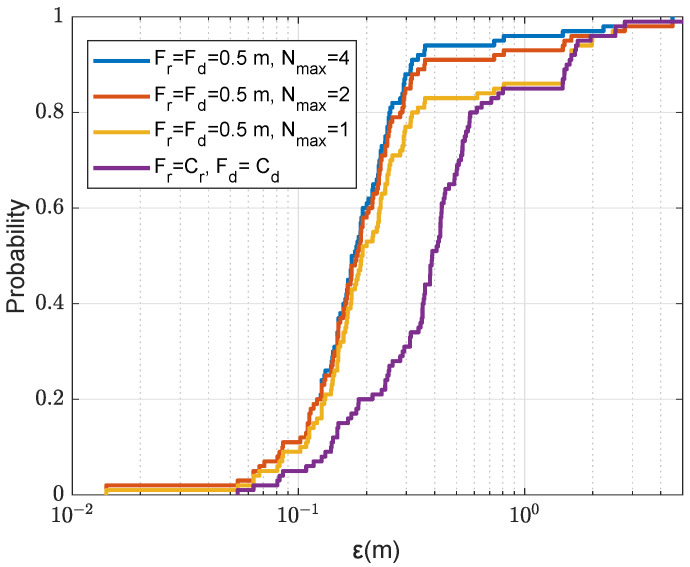
CDF for the localization error ε in the acoustic environment with the uniform SSP against the number $N_{\max}$ of refinement areas; $N_{T}=1$, the size of a refinement area is 2m×2m as shown in Figure 5a.

**Figure 13 sensors-22-06968-f013:**
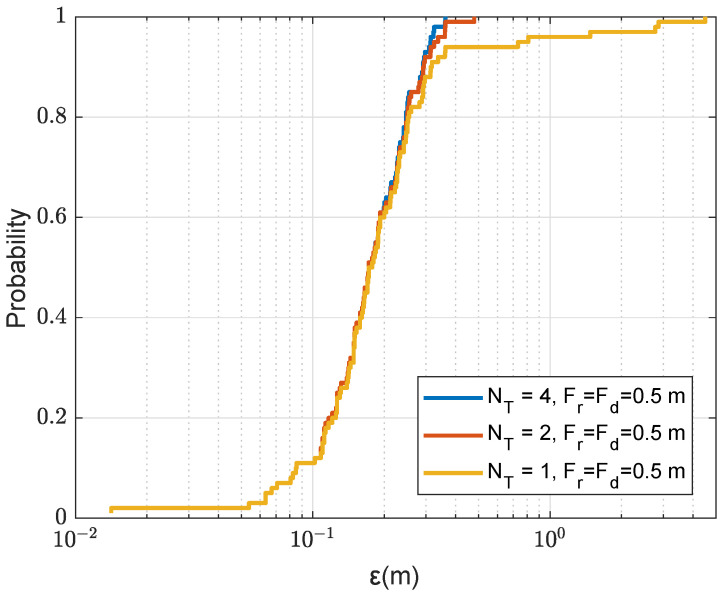
CDF for the localization error ε in the acoustic environment with the uniform SSP against the number of transmit antennas $N_{T}$; $F_{r}=F_{d}=0.5\;m$, the number of refinement areas is $N_{\max}=4$, where only one point is removed after finding the next maximum and the refinement area is 2m×2m as shown in Figure 5a.

**Figure 14 sensors-22-06968-f014:**
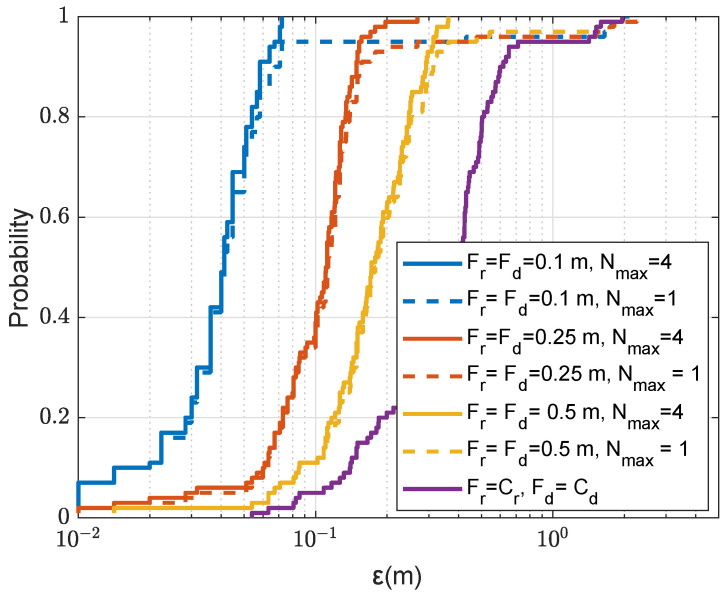
CDF for the localization error ε in the acoustic environment with the uniform SSP against the refinement step and multiple refinement areas, where only one point is removed after finding the next coarse AF maximum; $N_{T}=4$, the refinement area is 2m×2m as shown in Figure 5a.

**Figure 15 sensors-22-06968-f015:**
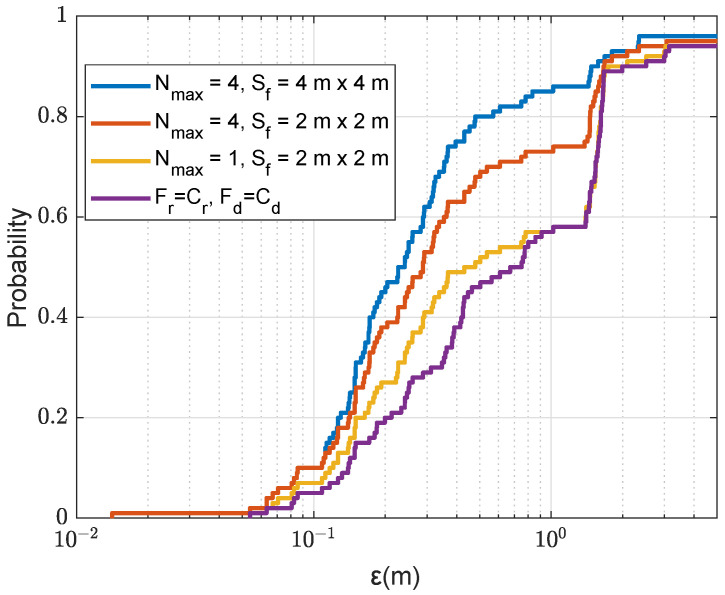
CDF for the localization error ε in the acoustic environment with the SWellEx-96 SSP against the size of the refinement area $S_{f}$ and number of refinement areas $N_{\max}$, where one points are removed after finding the next coarse AF maximum; $N_{T}=1$; $F_{r}=F_{d}=0.5$ m.

**Figure 16 sensors-22-06968-f016:**
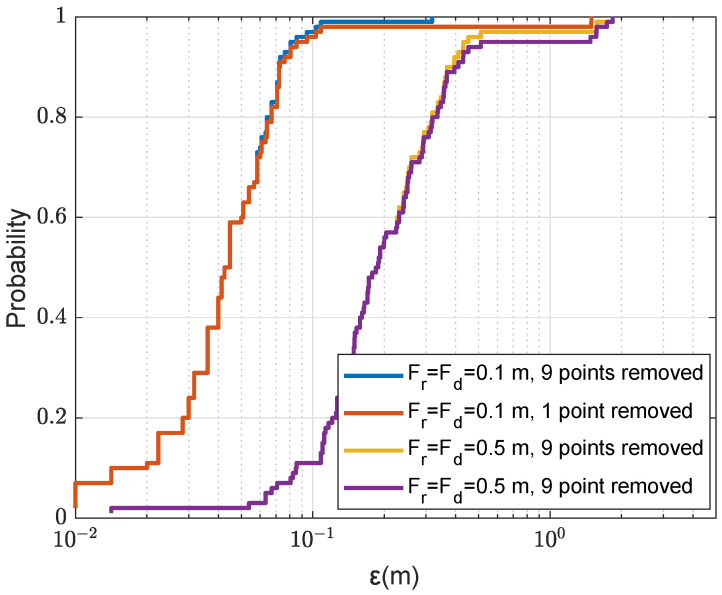
CDF for the localization error ε in the acoustic environment with the SWellEx-96 SSP against refined steps $F_{r}=F_{d}$; $N_{T}=4$; $N_{\max}=4$, after finding the AF maximum, two cases considered, one point and nine points are removed from the coarse grid map as described in Section 3.3; the refinement area is $S_{f}=4\;m\times 4\;m$ (as shown in Figure 5b).

**Figure 17 sensors-22-06968-f017:**
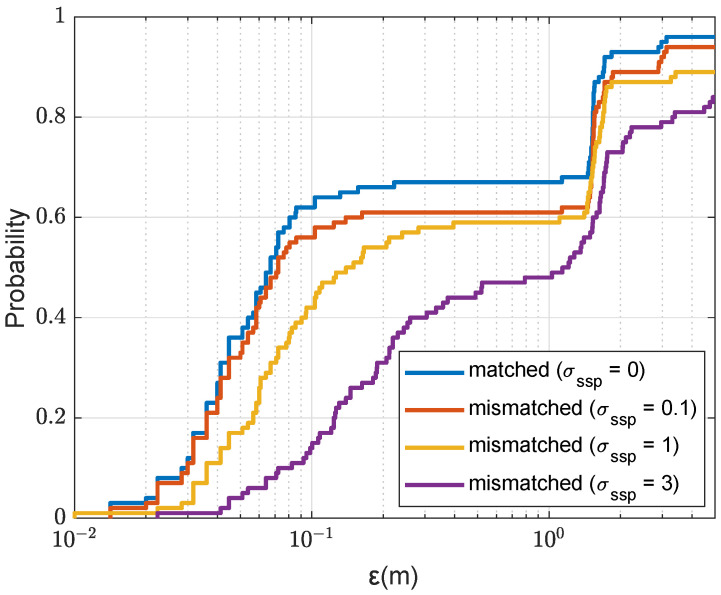
CDF for the localization error ε in mismatched acoustic environments against the variance sound speed $\sigma_{\mathrm{ssp}}^{2}$; $N_{T}=1$; $N_{\max}=4$, nine points are removed after finding the next coarse AF maximum; the refinement steps, $F_{r}=F_{d}=0.1\;m$; the size of a refinement area is 2m×2m (as shown in Figure 5a).

**Figure 18 sensors-22-06968-f018:**
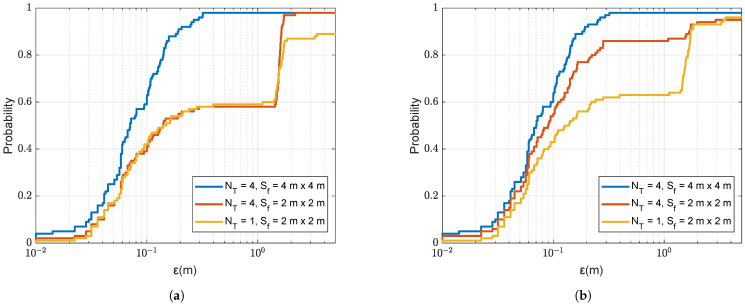
CDF for the localization error ε in a mismatched acoustic environment when $\sigma_{\mathrm{ssp}}=1\;m/s$ as shown in Figure 3: (**a**) Depth is unknown; (**b**) Depth is known. $N_{T}=4$; $N_{\max}=4$, nine points are removed after finding the next coarse AF maximum; the refinement steps, $F_{r}=F_{d}=0.1\;m$; the size of two refinement areas are 2m×2m and 4m×4m (as shown in Figure 5).

**Figure 19 sensors-22-06968-f019:**
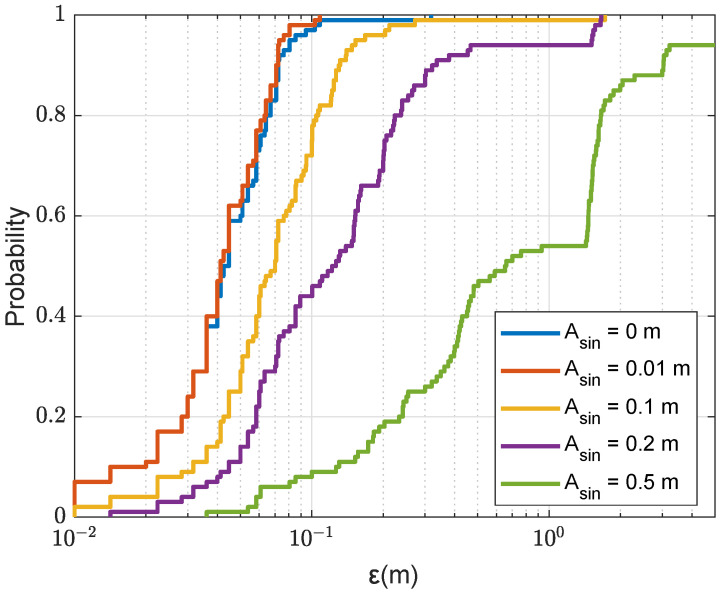
CDF for the localization error ε in the acoustic environment of the sinusoidal surface with the SWellEx-96 SSP against different amplitudes $A_{\sin}$ ($A_{\sin}=0$ indicates a flat surface); $N_{T}=4$; $N_{\max}=4$, nine points are removed after finding the next coarse AF maximum; the refinement steps, $F_{r}=F_{d}=0.1\;m$; the size of the refinement area is 4m×4m (as shown in Figure 5b).

**Figure 20 sensors-22-06968-f020:**
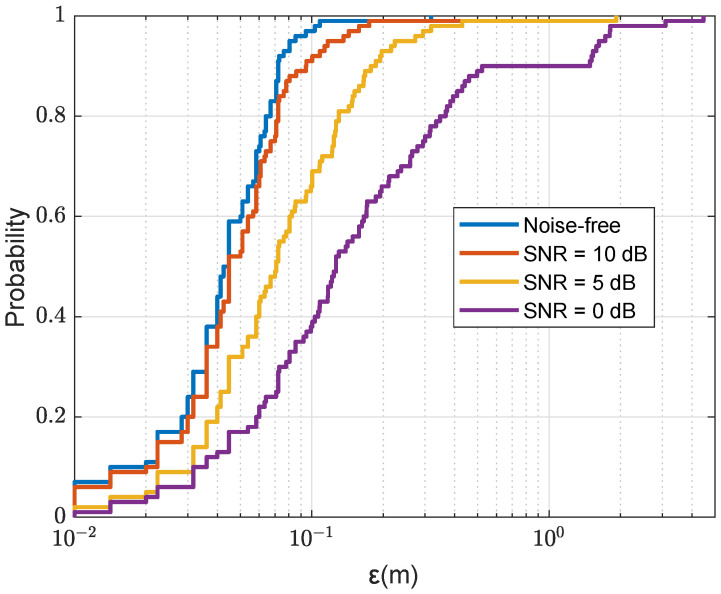
CDF for the localization error ε in the acoustic environment with the SWellEx-96 SSP against the SNR of channel response estimation; $N_{T}=4$; $N_{\max}=4$, nine points are removed after finding the next coarse AF maximum; the refinement steps, $F_{r}=F_{d}=0.1\;m$; the size of the refinement area is 4m×4m (as shown in Figure 5b).

**Table 1 sensors-22-06968-t001:** Simulation parameters used in an example of receiver localization.

Variable Name	Value	Description
*B*	1024Hz	Frequency bandwidth
$C_{d}$	1m	Coarse grid step in depth
$C_{r}$	1m	Coarse grid step in range
$D_{T}$	50,60,70,80m	Depth of transmit antennas
$D_{l}$	70 m	Depth for area of interest
$f_{c}$	3072Hz	Carrier frequency
*K*	1024	Number of subcarriers
$N_{T}$	4	Number of transmit antennas
$R_{l}$	120 m	Range for area of interest
δ	1Hz	Subcarrier spacing
τ	[−0.5s,0.5s]	Delay uncertainty interval

**Table 2 sensors-22-06968-t002:** Parameters for coarse receiver localization.

Variable Name	Value	Description
$C_{d}$	1m	Coarse grid step in depth
$C_{r}$	1m	Coarse grid step in range
$D_{T}$	50,60,70,80m	Depth of transmit antennas
$D_{l}$	200 m	Depth for area of interest
*K*	1024	Number of subcarriers
$N_{T}$	1, 2, 3, 4	Number of transmit antennas
$R_{l}$	500 m	Range for area of interest
$S_{c}$	201×501	Size of the coarse grid map
δ	1Hz	Subcarrier spacing
τ	[−0.5s,0.5s]	Delay uncertainty interval

## Data Availability

Not applicable.

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
