# Peer review of "Coarse-to-Fine Localization of Underwater Acoustic Communication Receivers"

_sensors, 2022, doi:10.3390/s22186968_

Round 1

Reviewer 1 Report

I highly appreciate the research work done by authors on matched field processing for localization of a single-antenna underwater acoustic communication receiver relative to one or more transmit antennas. The authors have proposed novel idea for localization in underwater acoustic channel using non-coherent AF. The have further proposed the location accuracy by refining methods. I suggest some changes in the manuscript which can increase the quality of the manuscript.

1.       In section 2, after line 94, However, there is an unknown propagation delay τ between the channel response estimate and channel responses in the grid map. Kindly elaborate how this delay is produced? How it is represented by factor e −j2π f τ in frequency domain? Kindly elaborate how equation 4 in derived for the measure for comparison of channel frequency responses.

2.       Kindly elaborate how the localization performance could be further improved by using the AF in eq 6.

3.       In line 112, kindly elaborate and give reason why the localization is improved by reducing the special sampling interval?

4.       At page 8, To compute amplitudes and delays for rays arriving at the point (x,y), we will be using the approach in [23]. It is suggested to describe the process of reference in a single paragraph referring the work of Siderius, M. and Porter, M.B.

5.       In section refinement and multiple refinement, kindly discuss the results of refinement achieved using this novel refinement technique.

6.       It is highly suggested that conclusion should be replaced by discussion. In this section, the authors should discuss the pros and cons of coherent AF, non-coherent AF, refinement and multiple refinement with multiple frequencies and SSP and done in numerical results. The authors should also present the comparison in a tabular form so it can be easy for readers to use the novel localization technique proposed in this work.

Author Response

Response to Reviewer 1

Thank you for the comments.
We are uploading (a) our point-by-point response to the comments (below) (response to reviewer), (b) an updated manuscript with blue highlighting indicating changes, and (c) a clean updated manuscript without highlights (Main Manuscript).

Best regards,
Pan He et al.
Department of Electronic Engineering
University of York

Reviewer 2 Report

  1. Is the manuscript
    1. clear,
    • The overall description is clear and easy to follow.
    • Section 3.4 specifically should be re-worked for clarity. Consider placing the underlying mathematical formulations in equation environments.
    1. relevant for the field and
    • The contribution to the field is clear and relevant.
    1. presented in a well-structured manner? 
    • The localization problem description in the introduction should be improved to be more explicit: Where is the information about the location required and where is the estimation taking place? In what way is your approach valid/helpful to solve that problem? What are limiting factors of your approach?
    • L.52ff: Rephrase to first descripe the first step, then the second step for clarity.
  2. Are the cited references
    1. mostly recent publications (within the last 5 years)
    • The references contain a good selection of recent works, but do not include any MDPI publications.
    1. and relevant?
  3. Does it include an excessive number of self-citations?
    • P.2: Reduce number of references to a single paper (by co-author, 7 references in one page). Consider separating the introduction, e.g., into "Relevant works of other groups" and "Prior works of own group", or similar.
    • Unclear, if manuscript is using concepts published in other prior publications of the group in this field, if so, please add those to the references to those, too.
  4. Is the manuscript
    1. scientifically sound?
    • L.74: How can you assume the environment is known and stable? Is this assumption based on the short duration of a localization operation or does your approach require prior characterization of the channel?
    • L.81: What happens if the assumption that the receiver is inside the area of interest does not hold?
    • Does the receiver node track the transmitting antennas by the signal content, transmission intervals or coding scheme, e.g., similar to GNSS's m-sequences?
    • What happens in the presence of strong reflectors, e.g., small UAV in front of large ship or near the surface or the ground? Do you assume a laterally endless body of water?
    • The self-localization in depth is quite accurate through a manometer, so why is this information not considered or used to reduce the degrees of freedom of this approach?
    • Fig.2: Why is there a general dependency of the coherent AF to range as visible in the background gradient from left to right? Is fading not considered in the channel model?
  5. Is the experimental design
    1. appropriate to test the hypothesis?
    • The simulations seem well designed to exhaust the model's capabilities, a real-world experimental verification still would help to improve the results and show potential limitations of the model.
  6. Are the manuscript’s results
    1. reproducible based on the details given in the methods section?
    • The manuscript explains the methodology well enough to be potentially reproducible, especially since the experimental design is restricted to simulations.
  7. Are the figures
    1. appropriate?
    2. Do they properly show the data?
    3. Are they easy to interpret and understand?
    • Fig.1: The image quality is rather low as there are several partially hidden lines and distorted circles. Can you redraw the image or export it in a format more suitable for print embedding? Also consider changing the color scheme, as yellow on white is very low in contrast.
    • The scales of the empirical cummulative distribution plots is rather incoherent, e.g., if you compare fig.11 with fig.20. For ease of comparing the general effects and improvements, it might be benefitial to settle on common scales for all eCDF plots.
  8. Are the tables
    1. appropriate?
    2. Do they properly show the data?
    • Tab.2: To improve the ease of parsing the information contained I suggest to separate the variable names from the values, place the names in the first column, list the values second, and the description last. The sorting of the table seems to be random and could be changed to increasing alphabetical by variable name, given the suggested changes above.
    1. Are they easy to interpret and understand?
  9. Are the images
    1. appropriate?
    2. Do they properly show the data?
    3. Are they easy to interpret and understand?
  10. Are the schemes
    1. appropriate?
    2. Do they properly show the data?
    3. Are they easy to interpret and understand?
  11. Is the data interpreted appropriately and consistently throughout the manuscript?
    1. Please include details regarding the statistical analysis or data acquired from specific databases.
    • The statistical analysis seems thorough and well proportioned.
  12. Are the conclusions
    1. consistent with the evidence and arguments presented?
    • The qualitative conclusion is fine, but the manuscript lacks a quantitative conclusion which clearly summarizes the improvements' effects in a numerical fashion.
  13. Is the ethics statement adequate?
    • The work lacks any consideration of the noise pollution introduced to the underwater acoustic channel and therefore the underwater habitat given the suggested frequency is used by most underwater animals in coastal regions and the distances suggest rather high power amplitudes inside the focus area of the beam. Therefore, the ethical statement does not seem appropriate.
  14. Is the data availability statement adequate?
    • The authors do not consider this applicable, which I do not follow. Consider offering the data for other groups to compare or verify your findings - at least on personal request.

Author Response

Response to reviewer 2:

Thank you for the comments.
We are uploading (a) our point-by-point response to the comments (below) (response to reviewers), (b) an updated manuscript with blue highlighting indicating changes, and (c) a clean updated manuscript without highlights (Main Manuscript).

Best regards,
Pan He et al.
Department of Electronic Engineering
University of York

Reviewer 3 Report

  1.  The variable name in the picture should be consistent with the text.

  2. The Non-coherent AF and refinement used in this paper do improve the localization accuracy, and compare the accuracy of the non-coherent AF and the coherent AF in the coarse localization phase, but should be compared with some other excellent localization algorithms in terms of efficiency and accuracy.

  3. Section 3.4 discusses the complexity part of the algorithm, whether we can add pictures and compare them with other algorithms, so that readers can feel more direct.

Author Response

Response to reviewer 3:

Thank you for the comments.
We are uploading (a) our point-by-point response to the comments (below) (response to reviewers), (b) an updated manuscript with blue highlighting indicating changes, and (c) a clean updated manuscript without highlights (Main Manuscript).

Best regards,
Pan He et al.
Department of Electronic Engineering
University of York

Reviewer 4 Report

Very well-designed and described paper with high overall merit. My suggestions are in the field of the description of statistics - I'm missing some small resumes after CDF computations. Also, some discussion is necessary at the end of the paper. The conclusions shall de extended showing the results for cases described in the article.

Author Response

Response to reviewer 4:

Thank you for the comments.
We are uploading (a) our point-by-point response to the comments (below) (response to reviewers), (b) an updated manuscript with blue highlighting indicating changes, and (c) a clean updated manuscript without highlights (Main Manuscript).

Best regards,
Pan He et al.
Department of Electronic Engineering
University of York

Reviewer 5 Report

1.     The authors propose and investigate techniques based on matched field processing for localization of a single-antenna UWA communication receiver relative to one or more transmit antennas.

2. The authors should provide a more detailed research motivation demonstration.

3.     In the figure 1, an example of two grid maps for a geographical area; every grid map corresponds to a specific transmit antenna should be demonstrated in detail.

4.     In the figure 3, sound speed profiles (SSPs): uniform and SWellEx-96 should be demonstrated in detail.

5.     The manuscript has 20 figures; the number of the figures should be decreased.

6.     Revise the English thoroughly before submission.

Author Response

Response to reviewer 5:

Thank you for the comments.
We are uploading (a) our point-by-point response to the comments (below) (response to reviewers), (b) an updated manuscript with blue highlighting indicating changes, and (c) a clean updated manuscript without highlights (Main Manuscript).

Best regards,
Pan He et al.
Department of Electronic Engineering
University of York

Round 2

Reviewer 1 Report

The authors have justified all the comments raised in the revision. There are few formatting errors.

There is a gap in several pages which needs to filled alligning the text and figures.

Author Response

Response to reviewer 1:

Thank you for the comments.

We have updated the manuscript by filling aligning the text and figures, attached with a clean updated manuscript without highlights.

Best regards,

Pan He et al.

Department of Electronic Engineering

University of York
